# Deep learning to predict rapid progression of Alzheimer's disease from pooled clinical trials: A retrospective study

Xiaotian Ma[1☯‡], Madison Shyer[2☯‡], Kristofer Harris[2], Dulin Wang[1], Yu-Chun Hsu[1], Christine Farrell[2], Nathan Goodwin[2], Sahar Anjum[2], Avram S. Bukhbinder[2,3], Sarah Dean[2], Tanveer Khan[2], David Hunter[2], Paul E. Schulz[2], Xiaoqian Jiang[1], Yejin Kim[1]*

**1** Department of Health Data Science and Artificial Intelligence, McWilliams School of Biomedical Informatics, The University of Texas Health Science Center at Houston, Houston, Texas, United States of America, **2** Department of Neurology, McGovern Medical School, The University of Texas Health Science Center at Houston, Houston, Texas, United States of America, **3** Division of Pediatric Neurology, Massachusetts General Hospital, Boston, Massachusetts, United States of America

☯ These authors contributed equally to this work.
‡ These authors share first authorship on this work.
* Yejin.Kim@uth.tmc.edu

**Data Availability Statement:** Data access restrictions apply. Investigators may contact Eli Lilly and Company for dataset licensing (https://www.lilly.com/clinical-research/clinical-trials) and

## Abstract

The rate of progression of Alzheimer's disease (AD) differs dramatically between patients. Identifying the most is critical because when their numbers differ between treated and control groups, it distorts the outcome, making it impossible to tell whether the treatment was beneficial. Much recent effort, then, has gone into identifying RPs. We pooled de-identified placebo-arm data of three randomized controlled trials (RCTs), EXPEDITION, EXPEDITION 2, and EXPEDITION 3, provided by Eli Lilly and Company. After processing, the data included 1603 mild-to-moderate AD patients with 80 weeks of longitudinal observations on neurocognitive health, brain volumes, and amyloid-beta (Aβ) levels. RPs were defined by changes in four neurocognitive/functional health measures. We built deep learning models using recurrent neural networks with attention mechanisms to predict RPs by week 80 based on varying observation periods from baseline (e.g., 12, 28 weeks). Feature importance scores for RP prediction were computed and temporal feature trajectories were compared between RPs and non-RPs. Our evaluation and analysis focused on models trained with 28 weeks of observation. The models achieved robust internal validation area under the receiver operating characteristic (AUROCs) ranging from 0.80 (95% CI 0.79–0.82) to 0.82 (0.81–0.83), and the area under the precision-recall curve (AUPRCs) from 0.34 (0.32–0.36) to 0.46 (0.44–0.49). External validation AUROCs ranged from 0.75 (0.70–0.81) to 0.83 (0.82–0.84) and AUPRCs from 0.27 (0.25–0.29) to 0.45 (0.43–0.48). Aβ plasma levels, regional brain volumetry, and neurocognitive health emerged as important factors for the model prediction. In addition, the trajectories were stratified between predicted RPs and non-RPs based on factors such as ventricular volumes and neurocognitive domains. Our findings will greatly aid clinical trialists in designing tests for new medications, representing a key step toward identifying effective new AD therapies.

request the data following the step-by-step process on the Vivli Platform (https://vivli.org/ourmember/lilly/). Code for building, training, and evaluating the models is available on the GitHub repo (https://github.com/xiaotian0328/AD-Rapid-Progression-Prediction). Specific frameworks and libraries we used are reported in S5 Table.

**Funding:** PS is funded by the McCord Family Professorship in Neurology, the Umphrey Family Professorship in Neurodegenerative Disorders, multiple NIH grants and several foundation grants (1R01AG080137-01A1, 1RF1AG072491-01, 1R03AG077191-01, 1U01AG079847-01A1, 5R01AG66749-03, 1R01AG083039-01, 5R01AG067498-03, 1RF1AG055053-0A1, 1R01AG062690-01, 1R01AG059321-01A1, 1R01DE07027, 1R01AG082721-01, 2124789, AGT002985, AGT008724, AGT009122), and contracts with multiple pharmaceutical companies related to the performance of clinical trials (ALZ-80-AD301-AGT005383, AGT004414, AGT003423, AGT003882, AGT006620, AGT005768, AGT004564, AGT009188, AGT006056, AGT006764, AGT008197, AGT010139, AGT011949). There is no perceived financial conflict between these activities and this manuscript. XJ is CPRIT Scholar in Cancer Research (RR180012), and he was supported in part by Christopher Sarofim Family Professorship, UT Stars award, UTHealth startup, the National Institute of Health (NIH) under award number R01AG066749, R01LM013712, R01LM014520, R01AG082721, R01AG066749, U01AG079847, U01TR002062, U01CA274576, and the National Science Foundation (NSF) #2124789.

**Competing interests:** PS is funded by the McCord Family Professorship in Neurology, the Umphrey Family Professorship in Neurodegenerative Disorders, multiple NIH grants and several foundation grants (1R01AG080137-01A1, 1RF1AG072491-01, 1R03AG077191-01, 1U01AG079847-01A1, 5R01AG66749-03, 1R01AG083039-01, 5R01AG067498-03, 1RF1AG055053-0A1, 1R01AG062690-01, 1R01AG059321-01A1, 1R01DE07027, 1R01AG082721-01, 2124789, AGT002985, AGT008724, AGT009122), and contracts with multiple pharmaceutical companies related to the performance of clinical trials (ALZ-80-AD301-AGT005383, AGT004414, AGT003423, AGT003882, AGT006620, AGT005768, AGT004564, AGT009188, AGT006056, AGT006764, AGT008197, AGT010139, AGT011949). He serves as a consultant and speaker for Eli Lilly, and Acadia Pharmaceuticals. No other authors have declarations to disclose.

## Author summary

Alzheimer's Disease (AD), a progressive brain disorder that affects memory and cognitive skills, has different rates of progression in different individuals. Identifying rapid progressors (RPs) is vital for conducting clinical trials and determining effective treatments. RPs may exhibit distinctive characteristics and underlying pathophysiological differences compared to non-RPs, making it essential to identify RPs in randomized control trials (RCTs) to ensure balance between placebo and treated groups, or even develop their own trials if necessary. In this study, we aimed to develop deep learning models to detect rapid AD symptom progression and identify features contributing most to the model prediction using placebo-arm RCT data. This prediction can help refine subject selection in clinical trials for AD treatment. Clinical trialists could identify rapidly progressing patients in current trials and separate them within the trial(s), similar to the approach for patients with the APOEε4 allele. It could also enable clinical trials to develop RP-specific therapeutic interventions.

## Introduction

Alzheimer's disease (AD) is the sixth leading cause of death in the United States, affecting six million Americans aged 65 and older [1]. Potential therapeutic targets have been sought, leading to various pharmaceutical interventions tested in randomized clinical trials (RCTs). Rapid progressors (RPs) are a subset of AD patients experiencing faster rates of cognitive decline than average AD patients within a defined period of time. Including RPs in RCTs presents challenges in accurately detecting treatment effects on the primary efficacy outcomes [2,3], potentially affecting AD clinical trials like EMERGE and ENGAGE, which investigated aducanumab [4]. Identifying RPs before planning clinical trials and creating tailored trials can enable focused therapy development.

Predicting rapid progression, however, is challenging, as no universally agreed definition exists [5,6]. For instance, the FDA recently recognized a definition for the CDR-SB that requires a change of over eight points within 18 months [7], but the justification is not well described.

Potential prognostic factors affecting the rate of progression include fluid biomarkers (e.g., amyloid-beta levels [8,9]), neurodegeneration (e.g., brain atrophy [10,11]), and cognitive impairment rate [12–14]. Various data-driven predictive models using different data sources (e.g., ADNI [8,9,11,13,15], CODR-AD [14], and independent cohorts of investigators [10,12]) and methodologies (e.g., clustering [11,13], dimensionality reduction [10], conditional restricted Boltzmann machine [14], gradient boosting decision tree [8], and deep learning model [15,16]) have achieved limited success in predicting AD patients' future cognitive scores, as seen in The Alzheimer's Disease Prediction Of Longitudinal Evolution (TADPOLE) [17] challenges.

The placebo arms of randomized clinical trial (RCT) data can complement prior studies of AD progression rates. In this study, we hypothesized that longitudinal changes during the initial months of observation could be used to extrapolate subsequent progression rates. We leveraged longitudinal observations from the placebo arms of three RCTs enrolling mild or mild-to-moderate AD patients (EXPEDITION [18], EXPEDITION 2 [18], and EXPEDITION 3 [19]) to predict RPs. We developed deep recurrent neural networks with attention mechanisms to flexibly learn from multimodal information and predicted rapid progression on four

commonly used tests of neurocognition and/or functional impairment: the 14-item cognitive subscale of the Alzheimer's Disease Assessment Scale (ADAS-Cog14) [20,21], the Alzheimer's Disease Cooperative Study Activities of Daily Living Inventory (ADCS-ADL) [22], the Clinical Dementia Rating Scale-Sum of Boxes (CDR-SB) [23,24], and the Mini-Mental State Examination (MMSE) [25]. We used early observations (initial 28 weeks based on available data) to predict RPs at week 80, analyzed important features, and plotted AD progressive trajectories to capture significant predictors aligned with clinical knowledge. Our work contributes to the field by (1) building models based on placebo-arm RCT data to predict four definitions of RPs; (2) accurately predicting RPs using early observations (28 weeks) in RCTs; (3) identifying the most salient features for our models to predict RPs; and, (4) stratifying AD progressive trajectories using data-driven methods.

## Materials and methods

### Ethical approval

This UTHealth Institutional Review Board and the Committee for the Protection of Human Subjects (CPHS) reviewed this study, classifying it as "non-human subjects research" due to the use of de-identified retrospective data. Therefore, the study was approved with a waiver of HIPAA authorization and informed consent waiver.

### Overview

We employed deep learning models to predict RPs at week 80 using pooled RCT data, including demographics, comorbidities, baseline conditions, and the initial observations from baseline (Fig 1). We performed internal and external validation, analyzed relative feature importance, and visualized the average changes of important variables between true-positive (RP) and true-negative (non-RP) subjects over time to evaluate the model and substantiate clinical insights. The code for building, training, and evaluating the models can be found on GitHub (https://github.com/xiaotian0328/AD-Rapid-Progression-Prediction).

### Data source

We analyzed de-identified placebo-arm data from three RCTs of solanezumab in 2,097 patients with mild or mild-to-moderate AD, i.e., EXPEDITION (NCT00905372, 2009–2012) [18], EXPEDITION 2 (NCT00904683, 2009–2012) [18], and EXPEDITION 3 (NCT01900665, 2013–2017) [19]. The trials were sponsored by Eli Lilly and Company. All three trials had similar eligibility criteria, requiring patients to be 55 years or older and meet the AD diagnostic criteria from the National Institute of Neurological and Communicative Disorders and Stroke–Alzheimer's Disease and Related Disorders Association [26]. Exclusions were based on Modified Hachinski Ischemia Scale scores > 4 or Geriatric Depression Scale scores > 6. EXPEDITION and EXPEDITION 2 involved mild-to-moderate AD patients (MMSE scores 16–26) with PET scans and lumbar punctures at various time points. CSF collection and amyloid PET were conducted in subsets of the participants in EXPEDITION and EXPEDITION 2, and neither was required for study eligibility. EXPEDITION 3 enrolled only patients with mild AD (MMSE scores 20–26) with evidence of amyloid pathology.

### Variables

Since the cause of AD is unclear and many possible factors may contribute to its development, we opted to include as many variables as relevant to predict RPs. We employed deep learning models to decide on significant variables instead of assuming a priori their importance. Hence,

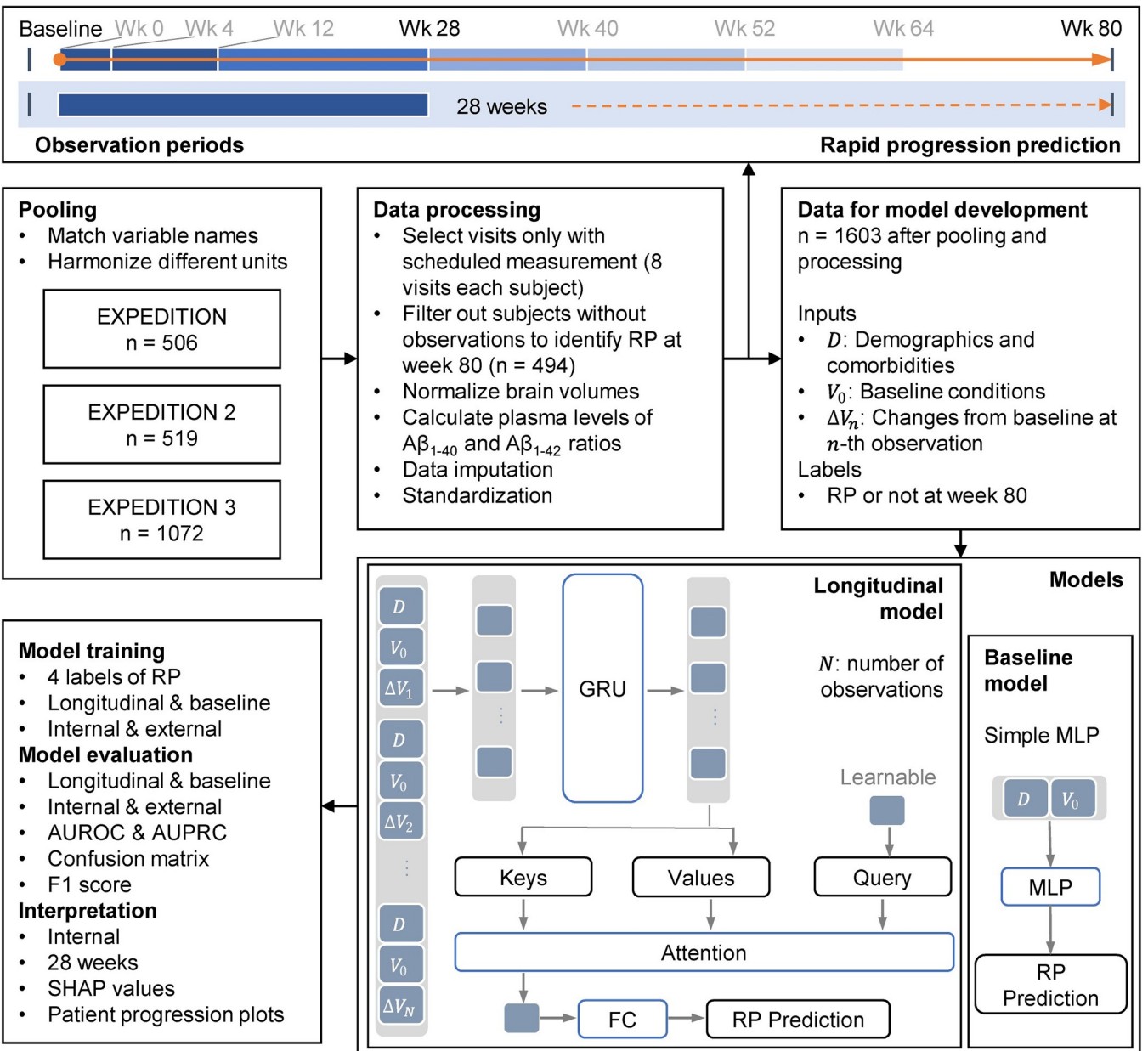

**Fig 1. Study overview for RP prediction.** Our cohort consisted of 2097 patients from 3 randomized control trials: EXPEDITION, EXPEDITION 2, and EXPEDITION 3. After pooling the trials and performing data processing, there were 1603 patients in the cohort. We compared the results of models using four different definitions of RP, which were based on changes in scores on ADAS-Cog14, ADCS-ADL, MMSE, and CDR-SB from baseline to week 80. We compared various observation periods from baseline up to week 80. We used the GRU with attention mechanism to learn the likelihood of RP from longitudinal and multimodal observations. After training and evaluating the models, we calculated feature importance to determine important modalities and features for prediction of RPs, and we plotted the trajectories of selected variables among RPs. CDR-SB = Clinical Dementia Rating Scale-Sum of Boxes; ADAS-Cog14 = 14-item cognitive subscale of the Alzheimer's Disease Assessment Scale; MMSE = Mini-Mental State Examination; ADCS-ADL = Alzheimer's Disease Cooperative Study–Activities of Daily Living Inventory; GRU = Gated Recurrent Unit; FC = Fully Connected layer; RP = Rapid Progression; SHAP = SHapley Additive exPlanations.

our models integrated 151 variables (S1 Table), such as demographics, comorbidities, neurocognitive measurements, imaging results, and amyloid-beta (Aβ) ratios. The baseline conditions included 22 demographic and 42 comorbid variables, while 87 variables had longitudinal measurements from baseline to 80 weeks. The longitudinal variables included (1) 73

neurocognition, neuropsychiatric symptom, and daily functioning measurements, (2) 6 brain regional volume measurements on MRI (e.g., volumes of hippocampi, entorhinal cortices, ventricles, and whole brain), (3) 1 amyloid PET imaging variable for the composite summary standard uptake value ratios (SUVRs), (4) 1 amyloid-beta (Aβ) plasma level ratio variable, and (5) 6 vital signs and anthropometry variables. The longitudinal variables contained both baseline measurements as well as the subsequent measurements at later weeks (i.e., weeks 4, 12, 28, 40, 52, 64, and 80). We aggregated components of the three neurocognitive tests into 6 cognitive domains: complex attention, executive function, language, learning and memory, perceptual-motor function, and social cognition (S2 Table).

## Data preprocessing

Since the three trials shared similar eligibility criteria and objectives, we pooled and preprocessed the clinical data collectively. We reviewed the respective protocols [18,19] and case report forms (CRFs) to confirm data collection similarities. Variable names were manually matched and different units were harmonized before longitudinal observations were processed.

Upon meeting the inclusion criteria, patients were enrolled and initial visits (week 0) were conducted. Variable values from the first encounters (screening or week 0) were set as baseline conditions, including time-variant features such as cognitive tests, MRI results, etc. Regular visits continued at least every four weeks for 80 weeks and we observed longitudinal biomarker changes in brain volumetry and plasma level $A\beta_{1-42}/A\beta_{1-40}$ ratio, as well as other markers such as cognitive/psychological scores. Longitudinal variable measurements were collected at weeks 0, 4, 12, 28, 40, 52, 64, and 80 (referred to as "observation periods" and cumulative; week 12 includes data from weeks 0, 4, and 12, week 28 includes weeks 0, 4, 12, and 28 data, etc.). The intervals between observation periods were prespecified in the structured protocols of the EXPEDITION trials from which our data were sourced. Such observation periods were designed with clinical relevance, as longer and later intervals are reflective of the expected gradual progression of AD. Hence, some time points were unavailable. We normalized the volumetric measurements of distinct brain regions by the total brain volume before applying a min-max normalization to the brain volumes within each trial to mitigate potential discrepancies arising from varying methodologies employed for the volume quantification [27]. For time-invariant variables (i.e., demographics, comorbidities, etc.), missing values were imputed using the Multiple Imputation by Chained Equation (MICE) [28] for demographics and baseline conditions and the most common class for comorbidities. Missing values in longitudinal variables were also imputed by MICE, assuming missing at random. We avoided temporal data leakage by imputing data within the observed periods according to the specific observation periods for each model. S1 Appendix and S1 Fig report details regarding how we evaluated and selected from several imputation strategies. Finally, we standardized numeric variables to zero mean and unit variance.

## Definitions of a rapid progressor

The proportion of patients with AD identified as "rapid progressors" in prior studies varies (10–30%) according to the definition used [5]. We defined rapid progressors (RPs) as the 10% of patients with the largest changes in cognitive scores from baseline to week 80. This was chosen empirically as clinical trialists have indicated that separating the fastest 10% of progressors would significantly benefit their trial design. We tested for RPs using these four cognitive measures: (1) the ADAS-Cog14 (0–90, higher scores indicating greater cognitive impairment) [20,21]; (2) the ADCS-ADL (0–78, lower scores indicating greater functional impairment)

[22]; (3) the CDR-SB (0–18, higher scores indicating greater functional impairment) [23,24]; and (4) the MMSE (0–30, lower scores indicating greater cognitive impairment) [25]. We used the Gaussian mixture model (GMM) (S1 Appendix, S2 Fig) to empirically validate the definition(s) and assess RP separation.

## Observation periods

We trained prediction models with data from the first $T$ weeks ($T$ = 4, 12, 28, 40, 52, 64) (Fig 1), comparing them to baseline measurement models. The models used varying observation periods and longitudinal variables (refer to "Data Preprocessing") to determine cognitive progression.

## Prediction model

We developed rapid progression prediction models using the first $N$ observations (corresponding to first $T$ weeks) of the longitudinal data, combining the gated recurrent unit (GRU) [29] and an attention mechanism for longitudinal inputs (Fig 1). The GRU is a gated mechanism used in recurrent neural networks with an update gate and a reset gate to control the flow of information from the previous hidden state and the current inputs. In addition, the attention mechanism can learn to assign larger weights to more important time steps, which would provide more information (S1 Appendix, S3 Fig). Demographics, comorbidities, baseline values, and changes from baseline formed the $n$-th sequential input. For comparison, we implemented simple two-layer multilayer perceptrons (MLPs) as baseline models using only demographics, comorbidities, and baseline measurements for the longitudinal variables. S1 Appendix provides more details about the models. We chose to train and test our models using the 28-week observation period as it is the earliest time point with significant data to predict RPs. This allows us to effectively capture early-stage trial data and accurately predict RPs at week 80.

## Evaluation

We evaluated the models using internal and external hold-out validation combined with 10-fold cross-validation. We split the pooled data (n = 1603) into a training set for 10-fold cross-validation (n = 1282, 80%) and a hold-out test set (n = 321, 20%). For external validation, we set aside the EXPEDITION 3 trial as the hold-out test set (n = 844), considering its different inclusion criteria, and treated the pooled EXPEDITION and EXPEDITION 2 trials (n = 759) as the training set for 10-fold cross-validation. We then split the training sets in both internal and external validation into 10 folds, reserving 1-fold for validation to select the best models and using the remaining 9 folds for model training each time.

We assessed prediction accuracy using the area under the receiver operating characteristic curve (AUROC) and the area under the precision and recall curve (AUPRC) (Fig 2). The AUROC measures the probability that the predicted scores for the ground-truth positive samples are larger than those of the negative samples, with 0.5 as the baseline referring to a random classifier. The AUPRC measures the trade-off between precision (fraction of true positive among predicted positive) and recall (fraction of true positive among ground-truth positive), with a higher AUPRC indicating the model can identify most of the positive samples without classifying too many negative samples as positive. The baseline value for AUPRC is the positive rate of the data, approximately 10% in our case. We reported the averaged AUROCs and AUPRCs over 10 folds for all trained models and calculated their 95% confidence intervals via $1.96 \times \text{SD}/\sqrt{n}$, where SD is the standard deviation of a metric over 10 folds, and $n = 10$ is the number of folds. To further ensure that our probabilistic predictions align well with the actual

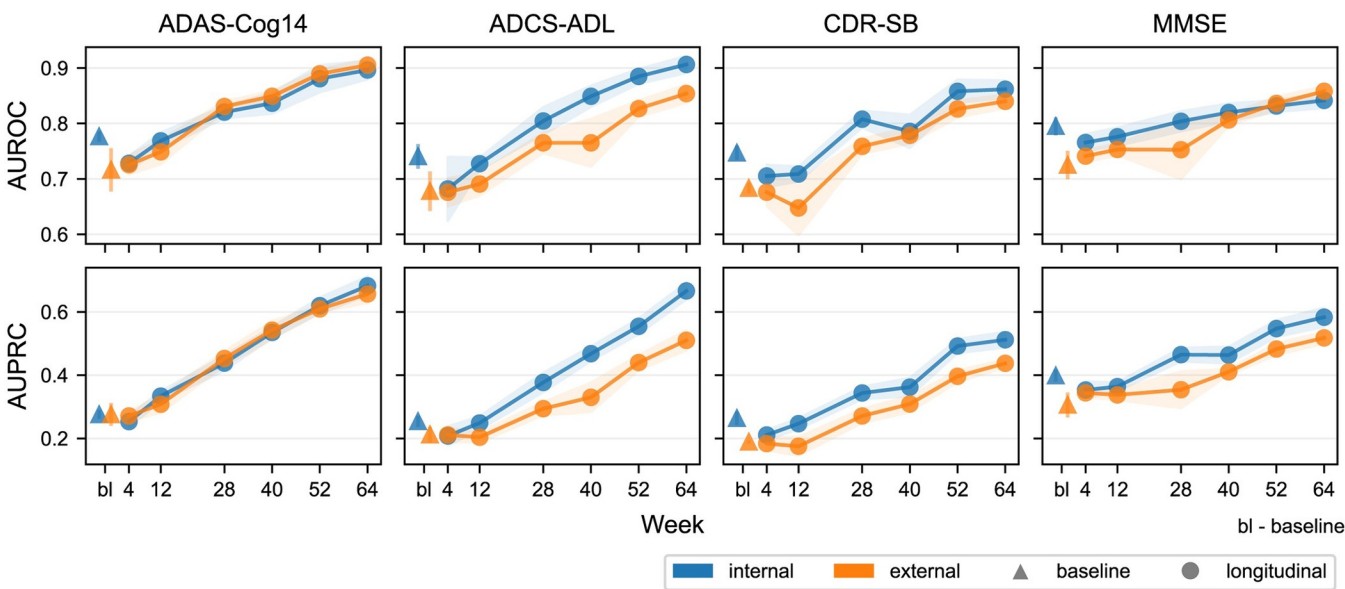

**Fig 2. AUROCs and AUPRCs of RP prediction models using different definitions and observation periods (including baselines).** Results include two validation modes: internal (to train and test with pooled data) and external (trained with EXPEDITION and EXPEDITION 2 and tested with EXPEDITION 3). We trained the models with 10-fold cross-validation and tested the trained models on the hold-out datasets for both evaluation settings and reported mean AUROCs and AUPRCs over the 10 folds. The 95% confidence intervals were calculated by $1.96 \times \text{SD}/\sqrt{n}$, where SD is the standard deviation over 10 folds and $n$ = 10 is the number of folds.

outcome distributions and that our models are reliable in clinical usage, we calibrated the model predictions and reported the calibration method and results in S1 Appendix and S4 Fig.

To further evaluate our model's binary prediction performances, we also displayed the confusion matrices and F1 scores for all models (Fig 3). The confusion matrix displays numbers of true negative (TN), false positive (FP), false negative (FN), and true positive (TP) in a 2×2 matrix, while the F1 score is the harmonic mean of precision and recall calculated as $F_1 = \frac{2 \cdot \text{precision} \cdot \text{recall}}{\text{precision} + \text{recall}} = \frac{2 \cdot \text{TP}}{2 \cdot \text{TP} + \text{FP} + \text{FN}}$. As we ran 10-fold cross-validation, we selected the classification thresholds for each fold to trade off the sensitivity and specificity on the validation set. Specifically, the thresholds were selected when sensitivity was larger than specificity with a margin ensuring a good sensitivity with a reasonable specificity (set to 0.15 in our case). Then, we had 10 binary predictions on the hold-out test set based on the selected thresholds. The final binary predictions were determined by the majority over the 10 predictions for all the models. The confusion matrices and F1 scores were computed based on the final predictions, with confusion matrices normalized over rows to show specificities in the upper left boxes of matrices and sensitivities in the lower right boxes of matrices (Fig 3).

## Interpretation

We calculated feature contributions to RP prediction using Shapley Additive Explanation (SHAP) [30] values (Fig 4, S1 Appendix) and grouped features into seven modalities: demographics, comorbidities and medications, plasma levels of Aβ, brain regional volumes, cognitive tests (ADAS-Cog14, CDR, MMSE), quality of life (ADCS-ADL, EQ-5D, QoL-AD), and neuropsychiatric symptoms (NPI). SHAP measures the importance score for each variable, indicating how much each variable contributes to the model's prediction for a specific instance. We reported the mean absolute SHAP values over 10 folds and calculated the 95% confidence intervals as we did in calculating AUROCs and AUPRCs. We also investigated

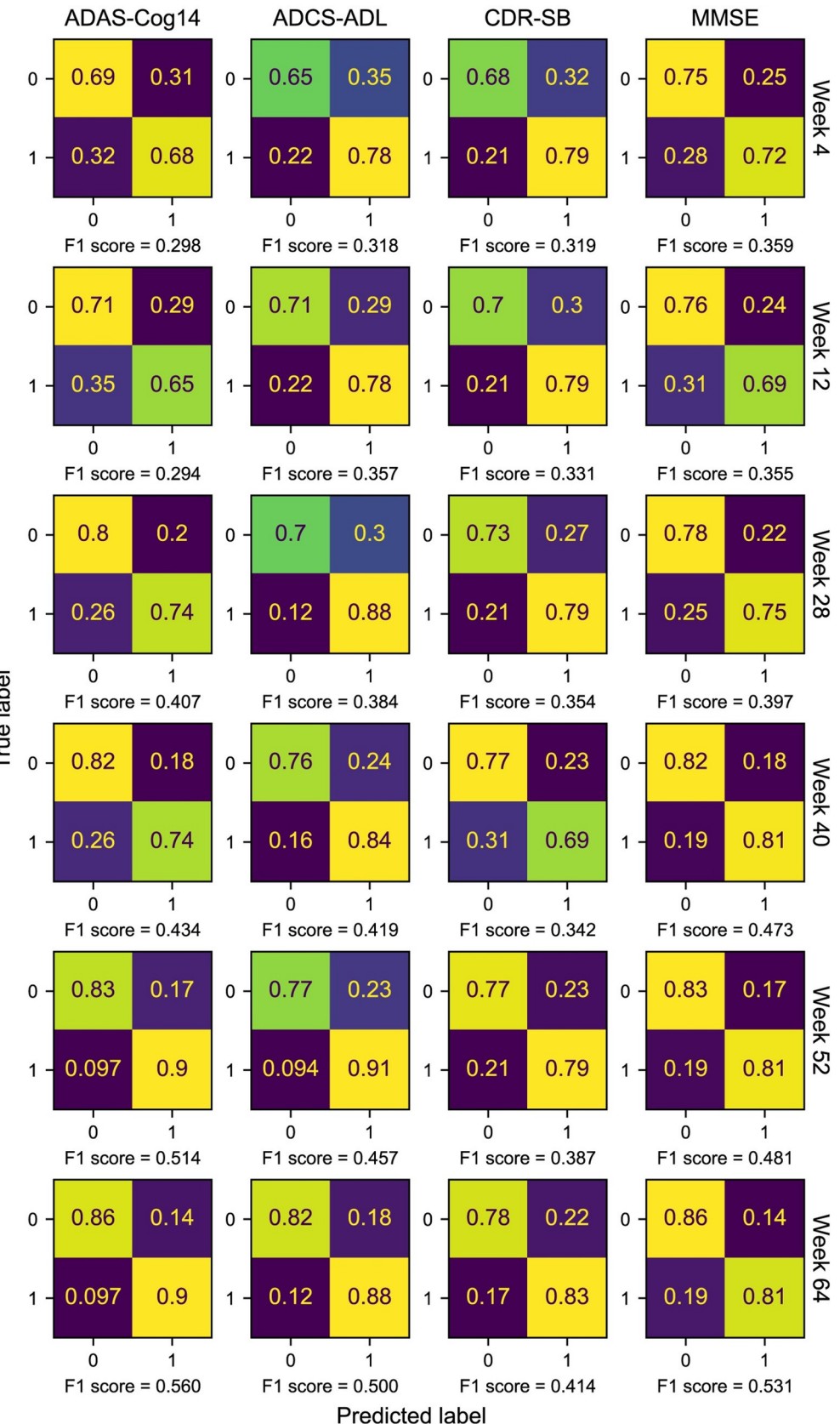

**Fig 3. Confusion matrices and F1 scores of RP prediction models using different definitions and observation periods.** Results include longitudinal models with internal validation. Confusion matrices are nomalized over rows, and corresponding F1 scores are reported under the matrices. Each normalized confusion matrix displays specificity in the upper left box and sensitivity in the lower right box.

modality- and feature-level interpretations using the 28-week period and visualized patient progression for important features, comparing predicted positive (RP) and predicted negative (non-RP) samples, with ground-truth positive and ground-truth negative as references (Fig 5, S1 Appendix). The thresholds to determine positive and negative samples were selected by the method described in the "Evaluation" section.

## Results

### Data

After preprocessing and removing 494 patients with missing observations, we finalized a pooled sample of 1,603 patients from EXPEDITION (n = 368), EXPEDITION 2 (n = 391), and EXPEDITION 3 (n = 844). Patient characteristics, lab values, and scans are in Table 1.

### Rapid progressors

As noted under Methods, RP thresholds were defined as the greatest 10th percentile of changes in cognitive scores from baseline to week 80, which were: ADAS-Cog14 total change > 22 (n = 147, 9.2%), ADCS-ADL total change < -24 (n = 152, 9.5%), CDR-SB total change > 6 (n = 150, 9.4%), or MMSE total change < -10 (n = 145, 9.0%) (S3 Table). After careful consideration of all relevant data per participant, 305 of 1603 patients were classified as rapid progressors by at least one of the four definitions. Using a Gaussian Mixture Model, the 10% thresholds stand on the tail of the major Gaussian component with a probability density close to 0 (S1 Fig).

### Rapid progressor prediction

Prediction accuracy increased with observation data duration from 4 to 64 weeks (Fig 2, S4 Table). Models using only baseline data had internal AUROCs ranging from 0.74 (95% CI 0.72–0.76) to 0.79 (0.78–0.81) and AUPRCs from 0.25 (0.24–0.27) to 0.40 (0.38–0.42); external AUROCs ranged from 0.68 (0.64–0.71) to 0.73 (0.70–0.75) and AUPRCs from 0.19 (0.18–0.19) to 0.31 (0.27–0.35).

Models using data up to week 28 outperformed baseline models with internal AUROCs from 0.80 (0.79–0.82) to 0.82 (0.81–0.83), internal AUPRCs from 0.34 (0.32–0.36) to 0.46 (0.44–0.49), external AUROCs from 0.75 (0.70–0.81) to 0.83 (0.82–0.84), and external AUPRCs from 0.27 (0.25–0.29) to 0.45 (0.43–0.48).

Prediction accuracies for the four RP definitions varied. Models predicting ADAS-Cog14--defined RPs using week 28 data had the highest AUROCs and better AUPRCs than other definitions. In contrast, models using only baseline data or data up to week 12 performed better when RP was defined by the change in the MMSE. However, observations up to weeks 4 or 12 did not provide enough information for an accurate prediction as AUROCs and AUPRCs were below a threshold deemed sufficient for clinical utility at these time points. Clinically, observations at week 40 and beyond are too late in the RCT to be meaningful. Therefore, the following analysis and discussion focus on observation periods up to 28 weeks.

In addition, our models reached F1 scores ranging from 0.29 to 0.56 overall on the internal hold-out test set, while the models using data up to week 28 had F1 scores from 0.35 to 0.41

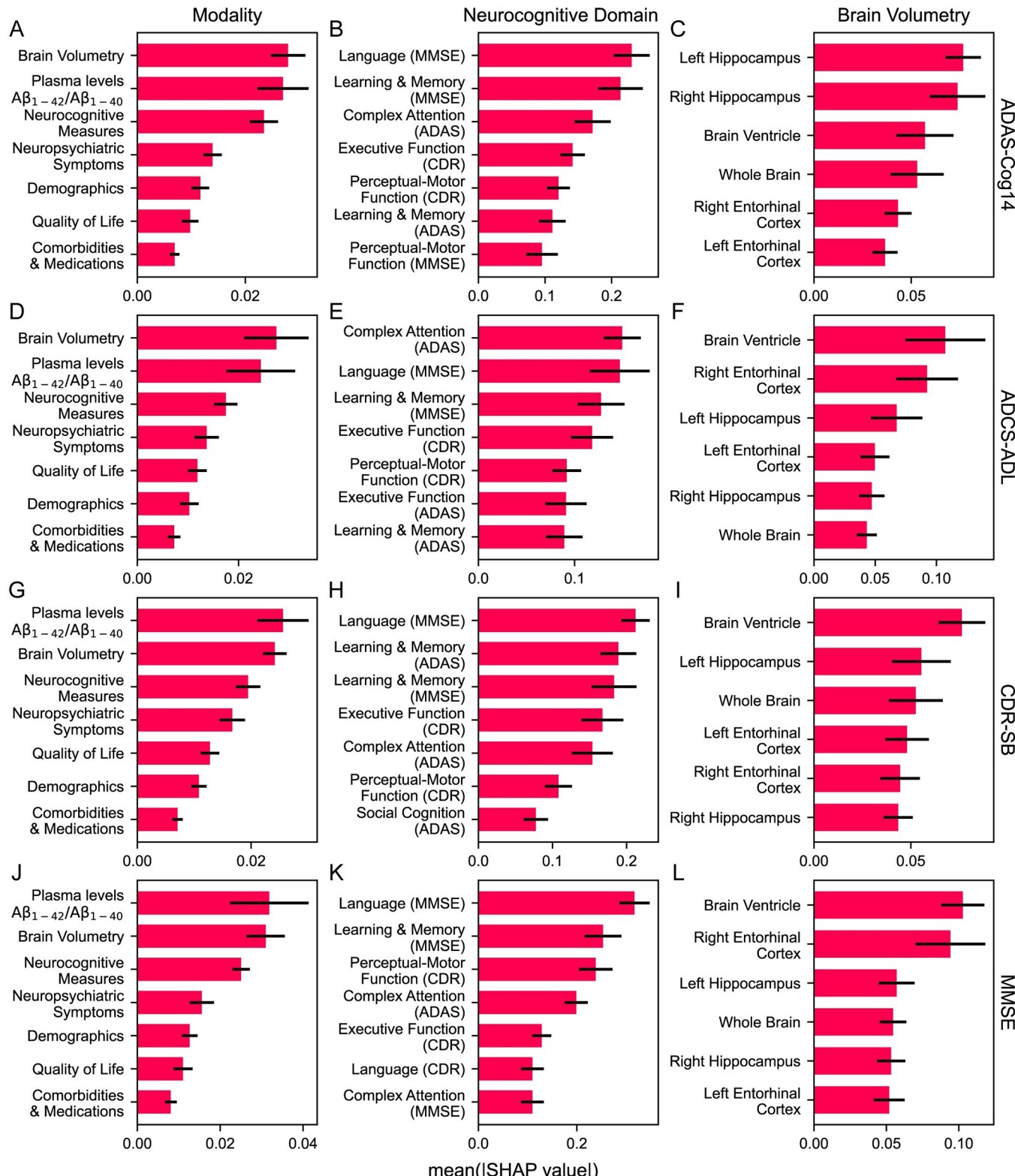

**Fig 4. Mean absolute SHAP values of important predictors contributed to models trained with 28-week observations.** The rows correspond to the four RP definitions: (A–C) ADAS-Cog14, (D–F) ADCS-ADL, (G–I) CDR-SB, and (J–L) MMSE. The columns correspond to three categories of features or feature groups: overall modalities (demographics, comorbidities and medications, neurocognitive measures, quality of life, neuropsychiatric symptoms, brain volumetry, and plasma level $A\beta_{1-42}/A\beta_{1-40}$ ratio), neurocognitive domains, and brain regional volumes. Due to the nature of the data, the x-axes in different

panels exhibit various ranges. We report mean SHAP values over the 10 folds. The 95% confidence intervals were calculated by $1.96 \times SD/\sqrt{n}$, where SD is the standard deviation over 10 folds and $n = 10$ is the number of folds.

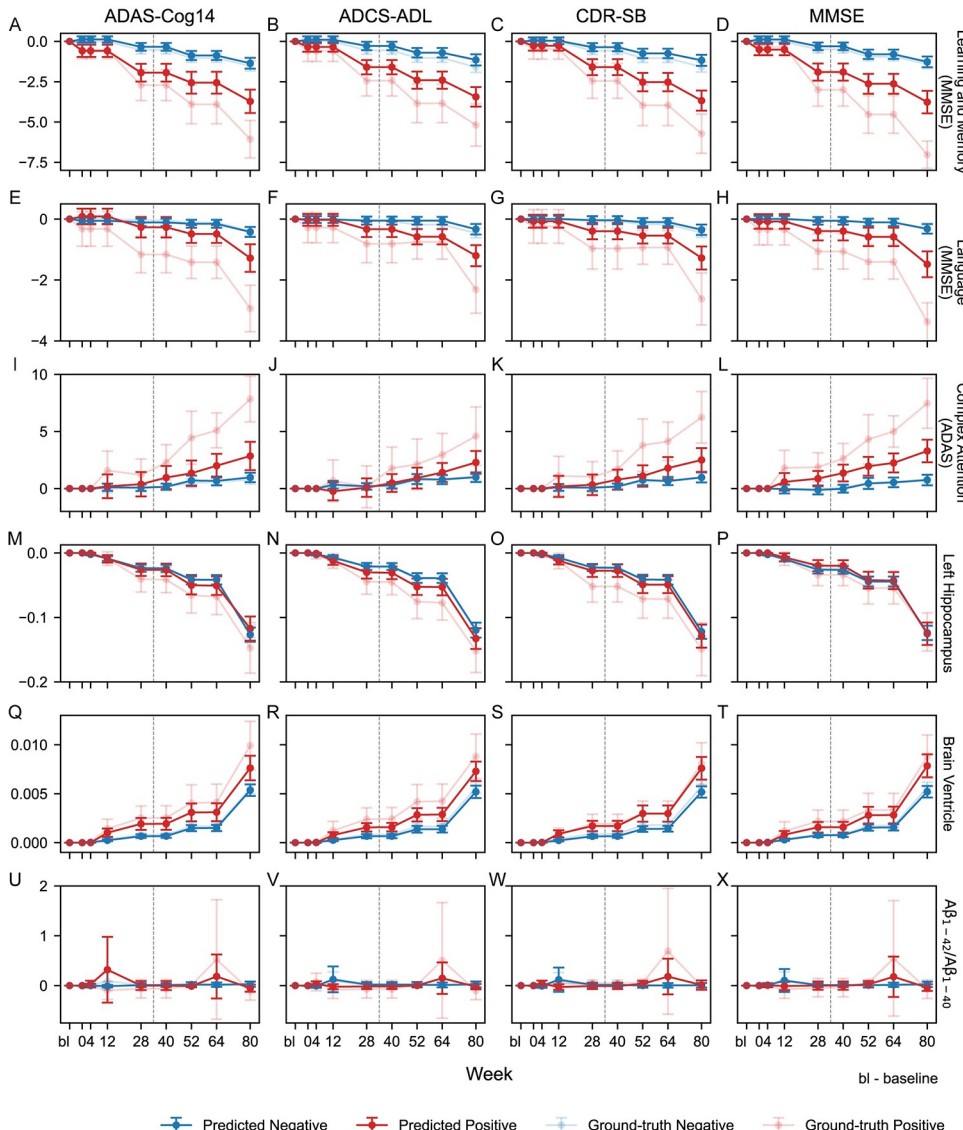

**Fig 5. Longitudinal distributions of important cognitive domains, relevant brain regional volumes, and plasma level Aβ$_{1-42}$/Aβ$_{1-40}$ ratio for each model among predicted positive (RP) and predicted negative (non-RP) subjects for models trained with 28-week observations.** The rows correspond to the important features: (A–D) learning and memory domain measured by MMSE, (E–H) language domain measured by MMSE, (I–L) complex attention domain measured by ADAS, (M–P) left hippocampus volume, (Q–T) brain ventricle volume, and (U–X) plasma level Aβ$_{1-42}$/Aβ$_{1-40}$ ratio. The columns correspond to the four RP definitions: ADAS-Cog14, ADCS-ADL, CDR-SB, and MMSE. The figure shows the average longitudinal changes of the selected variables from baseline to week 80 of predicted positive (RP, red) and predicted negative (non-RP, blue) subjects separated by trained models, respectively, on the internal test set. We also showed the ground-truth positive (light red) and ground-truth negative (light blue) for reference. The vertical reference line in each panel indicates the separation between observations used in training (left, up to 28 weeks) and those not used in training (right, more than 28 weeks).

**Table 1. Descriptive statistics of patient characteristics, lab values, and scans at baseline for three solanezumab trials.**

| | EXPEDITION | EXPEDITION 2 | EXPEDITION 3 | Pooled |
|---|---|---|---|---|
| Number of subjects | 368 | 391 | 844 | 1603 |
| **Demographics** | | | | |
| Age at baseline (years, mean (± SD)) | 74.5 (±7.8) | 71.8 (±7.7) | 72.8 (±7.7) | 73.0 (±7.8) |
| < 65 years old | 46 (12.5%) | 78 (19.9%) | 147 (17.4%) | 271 (16.9%) |
| ≥ 65 years old | 322 (87.5%) | 313 (80.1%) | 697 (82.6%) | 1332 (83.1%) |
| Age of symptom onset (years) | 70.0 (±8.0) | 67.5 (±8.2) | 68.5 (±7.8) | 68.6 (±8.0) |
| < 65 years old | 97 (26.4%) | 143 (36.6%) | 275 (32.6%) | 515 (32.1%) |
| ≥ 65 years old | 271 (73.6%) | 248 (63.4%) | 569 (67.4%) | 1088 (67.9%) |
| *Sex* | | | | |
| Female | 212 (57.6%) | 216 (55.2%) | 501 (59.4%) | 929 (58.0%) |
| Male | 156 (42.4%) | 175 (44.8%) | 343 (40.6%) | 674 (42.0%) |
| *Ethnicity* | | | | |
| Hispanic or Latino | 57 (15.5%) | 26 (6.6%) | 41 (4.9%) | 124 (7.7%) |
| Not Hispanic or Latino | 217 (59.0%) | 212 (54.2%) | 734 (87.0%) | 1163 (72.6%) |
| *Race* | | | | |
| White | 310 (84.2%) | 294 (75.2%) | 695 (82.3%) | 1299 (81.0%) |
| Black or African American | 13 (3.5%) | 1 (0.3%) | 13 (1.5%) | 27 (1.7%) |
| Asian | 42 (11.4%) | 96 (24.6%) | 65 (7.7%) | 203 (12.7%) |
| American Indian or Alaska Native | 1 (0.3%) | — | — | 1 (0.06%) |
| Multiple | 2 (0.5%) | — | 2 (0.2%) | 4 (0.2%) |
| *APOE type* | | | | |
| E2/E2 | 1 (0.3%) | 1 (0.3%) | — | 2 (0.1%) |
| E2/E3 | 16 (4.3%) | 9 (2.3%) | 23 (2.7%) | 48 (3.0%) |
| E2/E4 | 8 (2.2%) | 9 (2.3%) | 21 (2.5%) | 38 (2.4%) |
| E3/E3 | 102 (27.7%) | 126 (32.2%) | 248 (29.4%) | 476 (29.7%) |
| E3/E4 | 170 (46.1%) | 157 (40.2%) | 410 (48.6%) | 737 (45.9%) |
| E4/E4 | 45 (12.4%) | 52 (13.3%) | 121 (14.3%) | 218 (13.6%) |
| **Baseline conditions** | | | | |
| *Measures of neurocognitive and/or functional impairment* (mean (± SD)) | | | | |
| ADAS-Cog14 | 32.3 (±10.0) | 33.9 (±10.5) | 28.9 (±8.1) | 30.9 (±9.5) |
| ADCS-ADL | 62.5 (±10.9) | 61.5 (±12.0) | 67.4 (±8.5) | 64.9 (±10.4) |
| CDR-SB | 4.9 (±2.3) | 4.7 (±2.4) | 3.8 (±1.8) | 4.2 (±2.1) |
| MMSE | 21.3 (±3.0) | 21.2 (±3.1) | 23.0 (±2.0) | 22.2 (±2.7) |
| *Aβ Plasma levels* (median (Q1-Q3)) | | | | |
| $A\beta_{1-40}$ (pg/mL) | 188.6 (162.9–220.4) | 178.3 (154.0–204.7) | 228.1 (199.9–257.1) | 206.3 (173.0–240.7) |
| $A\beta_{1-42}$ (pg/mL) | 22.0 (22.0–45.2) | 22.0 (22.0–22.0) | 44.4 (22.0–55.1) | 22.0 (22.0–50.4) |
| $A\beta_{1-42}/A\beta_{1-40}$ ratio | 0.14 (0.11–0.22) | 0.14 (0.12–0.21) | 0.17 (0.11–0.23) | 0.15 (0.11–0.23) |
| PET composite summary SUVR (mean by whole cerebellum) (mean (± SD)) | 1.4 (±0.28) (76.6% missing) | 1.6 (±0.25) (87.7% missing) | 1.5 (±0.18) (14.6% missing) | 1.5 (±0.20) (46.7% missing) |
| *vMRI\** (mean (± SD)) | | | | |
| Whole brain volume ($cm^3$) | 1006.1 (±109.6) | 1008.8 (±107.3) | 975.6 (±101.8) | 990.7 (±106.1) |
| Ventricular volume ($cm^3$) | 49.4 (±24.6) | 44.8 (±21.9) | 48.8 (±21.9) | 48.0 (±22.6) |
| Left entorhinal cortex volume ($mm^3$) | 424.1 (±148.5) | 469.2 (±172.5) | 1375.8 (±381.8) | 1020.2 (±550.7) |
| Right entorhinal cortex volume ($mm^3$) | 419.4 (±167.0) | 434.9 (±160.6) | 1307.6 (±381.5) | 970.1 (±532.0) |
| Left hippocampus volume ($mm^3$) | 1714.1 (±382.4) | 1738.6 (±378.6) | 2950.0 (±530.9) | 2374.8 (±768.2) |
| Right hippocampus volume ($mm^3$) | 1759.5 (±386.5) | 1785.8 (±394.9) | 3065.3 (±564.2) | 2457.7 (±810.3) |

\* Not normalized by whole brain volumes

(Fig 3). The confusion matrices of 28-week models show sensitivities ranging from 0.74 to 0.88, with specificities ranging from 0.7 to 0.8, indicating good stratification between RPs and non-RPs.

We visualized the SHAP values by modalities and important variables within these modalities that contribute to the models' prediction using week 28 data (Fig 4). The most important modalities were the plasma $A\beta_{1\text{-}42}/A\beta_{1\text{-}40}$ ratio, brain volumetry, and neurocognitive measures. Comorbidities and medications were the least important. Dysfunction in important cognitive domains such as complex attention, learning and memory, and language contributed to the prediction of RPs (Fig 4B, 4E, 4H, 4K), with most of the important functions being those measured by MMSE. Brain ventricle volume and left hippocampal volume were among the top three most important imaging biomarkers in all the models (Fig 4C, 4F, 4I, 4L), while brain ventricle volume was the most important volumetric feature for ADCS-ADL, CDR-SB, and MMSE models.

We selected the three most common and important cognitive domains (complex attention calculated by ADAS-Cog14, language calculated by MMSE, and learning and memory calculated by MMSE), the brain volumetry measurements (left hippocampus and brain ventricle), and the plasma $A\beta_{1\text{-}42}/A\beta_{1\text{-}40}$ ratio for each model and reported the mean and confidence intervals of the longitudinal changes in those features from baseline to Week 80 (Fig 5). We focused on the divergence between predicted positives (predicted by the model as RP) and predicted negatives (predicted by the model as non-RP), with ground-truth positives and ground-truth negatives as references. Fig 5, then, shows an ideal match between predicted negative and ground-truth negative groups. However, the predicted positive groups are generally underfitted to the ground-truth positive groups. Nonetheless, Fig 5 still displays overall clear stratification between predicted positive and negative groups.

## Discussion

We developed models to identify rapid progressors (RPs) using pooled data from completed Phase III RCTs of mild-to-moderate AD. The purpose of these models was to predict rapid symptom progression in patients. We found that the plasma $A\beta_{1\text{-}42}/A\beta_{1\text{-}40}$ ratio, brain volumetry, and specific neurocognitive measurements were the most important features within and across models. Our models highlight key features that support previous evidence.

Our longitudinal models achieved higher accuracies with longer observation periods. The 12-week model had similar accuracy to the baseline model, suggesting that 12 weeks might be insufficient for identifying RPs. Longitudinal models with 28-week or longer observation periods had higher accuracies, with the potential for even greater accuracy with more observations. We also observed the magnitude of variable changes capturing the progression better than absolute values at any given time point [31].

Figs 4 and 5 show that the three most important predictors of RP were plasma $A\beta_{1\text{-}42}/A\beta_{1\text{-}40}$ ratio, brain volumetry, and specific cognitive decline areas (e.g., complex attention, learning and memory, language), corroborating the National Institute on Aging- Alzheimer's Association's ATN (amyloid, tau, and neurodegeneration) classification [32]. As other studies have shown, plasma levels of $A\beta$ are correlated with the presence of CSF $A\beta$ [33], and the $A\beta_{1\text{-}42}/A\beta_{1\text{-}40}$ plasma level ratio is a robust measurement that correlates with the presence of amyloid plaques–an important indicator for AD diagnosis [34]. The clinical presentation of AD progression is reflected in cognitive decline and the brain atrophy reflects neurodegeneration, which is indicated by changes of brain volumetry [35].

Plasma $A\beta$ levels had an impact on predicting RPs (Fig 4A, 4D, 4G, 4J) with predicted positive RPs showing increased fluctuation in longitudinal progression of plasma level $A\beta_{1\text{-}42}/A\beta_{1\text{-}40}$ ratio

(Fig 5U–5X). This indicates its potential utility in differentiating rapidly progressing AD patients from those with a slower progression.

An additional biomarker identified by the models as important for RP prediction was changes in the volume of the hippocampi, atrophy of which is closely associated with cognitive impairment [36]. Furthermore, the most important brain volume predictor for RP in ADCS-ADL, CDR-SB, and MMSE models was the trajectory of ventricular volumes, which is consistent with the compensatory enlargement of the ventricles caused by the atrophy and volume loss of brain tissue in neurodegenerative disorders such as AD (i.e., hydrocephalus ex vacuo) [37]. The trajectories of brain volumetry (Fig 5M–5T) were suggestive of slightly lower hippocampal volumes and greater ventricular volumes among predicted positive RPs compared to predicted negative RPs, but no significant divergence between these groups was observed (even between groups of ground-truth positive and ground-truth negative RPs), which warrants further investigation.

Lastly, we evaluated the importance of neurocognitive domains for the four cognitive test definitions (Fig 4B, 4E, 4H, 4K) and aimed to identify the most crucial domains and tests for early detection of RPs. A comprehensive analysis of these domains and tests will be presented in a follow-up study.

Overall, AD progression prediction has been challenging. The TADPOLE challenge [17], a community effort using Alzheimer's Disease Neuroimaging Initiative (ADNI) data, aimed to develop a model predicting AD progression; however, none of the algorithms proved to be sufficient for prediction. A recent publication [38] employed machine learning techniques on ADNI and clinical trial datasets to predict longitudinal progression using the CDR-SB; however, the CDR is more labor-intensive than other cognitive tests. Our study shows that the MMSE is, for the most part, more consistent in AUROCs and AUPRCs and supports the assertion that other tests or even a combination of tests may be more sensitive at detecting and predicting progression.

Our study also developed an attention-based GRU that integrated a wide range of multimodal temporal data and harnessed the advantages of pooled RCT data to achieve better AUROCs and AUPRCs, providing a framework for clinicians and clinical trialists to screen patients for appropriate drug trials. Moreover, by identifying RPs and distinguishing them from non-RPs, we can enhance the balance between placebo and treated groups, allowing for more accurate assessments of treatment efficacy and potentially improving the development of targeted interventions for this unique patient population.

Limitations of this study include the inability to consider certain biological variables due to the complexities and uncertainties surrounding AD's pathoetiologies [5], as well as defining RPs based on clinical neuro-psych tests and structural imaging instead of biological or neuro-cognitive evidence. Differences in the clinical trial inclusion and exclusion criteria could also influence this study population. For example, EXPEDITION and EXPEDITION 2 did not require CSF collection or amyloid PETs for study eligibility, and as a result, some patients may have had other etiologies for their dementia. This is important because we do not rely on population statistics and trends but rather on individual patient-level RCT data. Additionally, the systematic difference in data distribution by sources may limit the generalizability of our model to other RCTs with similar inclusion and exclusion criteria. Future studies will need to utilize a dataset based on clinical (e.g., EMR) rather than RCT data. It is possible that clinical trials may have a batch effect based on eligibility criteria and study sites.

To mitigate reproducibility issues, we performed external validation (with data from EXPEDITION 3 as the external hold-out test set). Fortunately, the analysis we used in our model is intended to capture components of standard protocols for clinical trials in order to ensure generalizability to other clinical trials' data. The main objective of this study was to develop a

predictive model using the clinical trial gold standard of available data—placebo arms—that can be extrapolated to data sources outside RCTs.

Furthermore, we propose potential future directions that are worth exploring. (1) Survival analysis could be explored by treating RP as a time-to-event variable, implementing a dynamic prediction model that can be updated over time [39], and evaluating the models by time-dependent AUROC. (2) More imputation methods could be explored on our baseline and longitudinal data, such as the data-driven missing value imputation approach [40], HyperImpute [41], or integrating the data imputation module into the prediction models [16]. (3) Investigations could be conducted by using adapted supervised machine learning algorithms [42,43] and comparing them to our neural network models in terms of performances and efficiency for longitudinal classification. (4) There is the potential for the model to be tuned in the future to include and assess other genetic components that are predictive of a rapid progressor early on [44,45]. (5) To capture more variations of the data, we could include more cohorts of AD trials or clinical data (i.e., ADNI datasets) as our internal and external validation, as well as implement multi-modal, multi-task deep learning models on different cohorts. Additionally, more advanced deep learning architectures, such as autoencoders and transformers, could be used to capture inherent variations of the tabular data.

## Conclusion

Our research represents a breakthrough in predicting the rapid progression of AD. By analyzing pooled placebo-arm data from three randomized controlled trials, we developed advanced deep learning models capable of accurately identifying rapidly progressive AD cases.

A critical discovery of our research was the identification of key biomarkers and neurocognitive health indicators that are instrumental in predicting RPs. These include the plasma level $A\beta_{1-42}/A\beta_{1-40}$ ratio, regional brain volumetry, and specific areas of cognitive decline. These findings are important in the context of AD therapy development, as they might empower researchers to design clinical trials that more effectively account for the diverse ways in which AD symptoms can manifest and progress in patients.

In summary, our research represents a significant advancement in understanding and predicting the rapid progression of AD. The utilization of deep learning models derived from comprehensive data across multiple trials has enabled us to identify crucial factors in AD progression. This enhances our ability to predict which patients are more likely to experience rapid progression. The implications of these findings are profound, paving the way for more targeted and effective clinical trials. This, in turn, promotes the development of AD therapies that are better suited to address the varied trajectories of symptom progression in patients, ultimately leading to more effective treatment strategies for this complex and very distressing disorder.

## Supporting information

**S1 Appendix. Additional materials, methods, and results.**
(DOCX)

**S1 Fig. AUROCs and AUPRCs of RP prediction models of different imputation strategies.**
(A) Baseline variables: mean imputation. Longitudinal variables: 1) last observation carried forward (LOCF); 2) linear imputation; and, 3) multiple imputation by chained equation (MICE). (B) Baseline variables: MICE. Longitudinal variables: 1) LOCF; 2) linear imputation; and, 3) MICE. (C) Baseline variables: 1) mean imputation; and, 2) MICE. Longitudinal variables: MICE.
(DOCX)

**S2 Fig. GMM fitting on the ADAS-Cog14, ADCS-ADL, CDR-SB, and MMSE to determine the four rapid progressor definitions.** The red dash lines indicate the various thresholds. The probability densities of the major Gaussian distribution at the thresholds of CDR-SB, ADAS--Cog14, MMSE, and ADCS-ADL are 0.00094, 0.00041, 0.0014, and 0.0023, respectively, which are close to zero, indicating that the thresholds selected can separate samples within the major distribution and those within the tails. The corresponding areas of tails cut by the thresholds are 0.0019, 0.00077, 0.00061, and 0.0022, respectively.
(DOCX)

**S3 Fig. Attention heat maps on observed time steps for all models.**
(DOCX)

**S4 Fig. Calibration curves for all calibrated models.** ECE: Expected Calibration Error; BS: Brier Score.
(DOCX)

**S1 Table. All variables used in the model training.**
(DOCX)

**S2 Table. Mapping between variables and neurocognitive domains.**
(DOCX)

**S3 Table. Number and percentage of rapid progressors separated by the four rapid progressor definitions.**
(DOCX)

**S4 Table. Results of AUROC and AUPRC on the internal and external test datasets.**
(DOCX)

**S5 Table. Frameworks and libraries used for building, training, and evaluating the models.**
(DOCX)

**S6 Table. TRIPOD checklist of this study.**
(DOCX)

## Author Contributions

**Conceptualization:** Xiaotian Ma, Madison Shyer, Kristofer Harris, Dulin Wang, Christine Farrell, Nathan Goodwin, Paul E. Schulz, Xiaoqian Jiang, Yejin Kim.

**Data curation:** Xiaotian Ma, Madison Shyer, Kristofer Harris, Dulin Wang, Yu-Chun Hsu, Xiaoqian Jiang, Yejin Kim.

**Formal analysis:** Xiaotian Ma, Madison Shyer, Kristofer Harris, Dulin Wang, Yu-Chun Hsu, Xiaoqian Jiang, Yejin Kim.

**Investigation:** Xiaotian Ma, Madison Shyer, Kristofer Harris, Dulin Wang, Christine Farrell, Nathan Goodwin, Sahar Anjum, Avram S. Bukhbinder, Sarah Dean, Tanveer Khan, David Hunter, Paul E. Schulz, Xiaoqian Jiang, Yejin Kim.

**Methodology:** Xiaotian Ma, Dulin Wang, Yu-Chun Hsu, Xiaoqian Jiang, Yejin Kim.

**Supervision:** Paul E. Schulz, Xiaoqian Jiang, Yejin Kim.

**Validation:** Xiaotian Ma, Dulin Wang, Yu-Chun Hsu, Xiaoqian Jiang, Yejin Kim.

**Visualization:** Xiaotian Ma, Yu-Chun Hsu.

**Writing – original draft:** Xiaotian Ma, Madison Shyer, Kristofer Harris.

**Writing – review & editing:** Xiaotian Ma, Madison Shyer, Kristofer Harris, Dulin Wang, Yu-Chun Hsu, Christine Farrell, Nathan Goodwin, Sahar Anjum, Avram S. Bukhbinder, Sarah Dean, Tanveer Khan, David Hunter, Paul E. Schulz, Xiaoqian Jiang, Yejin Kim.

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
