## [Decision Letter · Decision Letter 0]

21 Sep 2023

PDIG-D-23-00271

Machine learning to predict rapid progression of Alzheimer’s disease from pooled clinical trials: A retrospective study

PLOS Digital Health

Dear Dr. Ma,

Thank you for submitting your manuscript to PLOS Digital Health. After careful consideration, we feel that it has merit but does not fully meet PLOS Digital Health's publication criteria as it currently stands. Therefore, we invite you to submit a revised version of the manuscript that addresses the points raised during the review process.

Please submit your revised manuscript within 60 days Nov 20 2023 11:59PM. If you will need more time than this to complete your revisions, please reply to this message or contact the journal office at digitalhealth@plos.org. Please include the following items when submitting your revised manuscript:

We look forward to receiving your revised manuscript.

Kind regards,

Hualou Liang

Academic Editor

PLOS Digital Health

Journal Requirements:

1. We ask that a manuscript source file is provided at Revision. Please upload your manuscript file as a .doc, .docx, .rtf or .tex.

Additional Editor Comments (if provided):

Reviewers' comments:

Reviewer's Responses to Questions

**Comments to the Author**

1. Does this manuscript meet PLOS Digital Health’s publication criteria? Is the manuscript technically sound, and do the data support the conclusions? The manuscript must describe methodologically and ethically rigorous research with conclusions that are appropriately drawn based on the data presented.

Reviewer #1: Yes

Reviewer #2: Partly

Reviewer #3: Yes

Reviewer #4: Yes

2. Has the statistical analysis been performed appropriately and rigorously?

Reviewer #1: Yes

Reviewer #2: No

Reviewer #3: No

Reviewer #4: Yes

3. Have the authors made all data underlying the findings in their manuscript fully available (please refer to the Data Availability Statement at the start of the manuscript PDF file)?

Reviewer #1: Yes

Reviewer #2: No

Reviewer #3: Yes

Reviewer #4: Yes

4. Is the manuscript presented in an intelligible fashion and written in standard English?

Reviewer #1: Yes

Reviewer #2: Yes

Reviewer #3: Yes

Reviewer #4: Yes

5. Review Comments to the Author

Reviewer #1: Dear Author,

Your manuscript is well written and results are well supported by data and conclusion however

1. the abstract is unstructured it needs re-synthesis and should be structured.

2. the 'Author Summary' should be replaced by 'Summary'. the introduction and discussion needs to be more explanatory

Reviewer #2: The article entitled “Machine learning to predict rapid progression of Alzheimer’s disease from pooled clinical trials: A retrospective study” by Ma et al, developed deep learning models for predicting rapid progressors (RPs) patients of Alzheimer’s disease. They utilized the pooled de-identified placebo-arm data (n=1603) from 3 randomized controlled trials for building the models. The data consist of 80 weeks longitudinal study with a total of 151 variables belonging to five different data types i.e. demographics, comorbidities, neurocognitive measurements, imaging results, and amyloid-beta (Aβ) ratios. The model architecture consists of a gated recurrent unit (GRU) with an attention mechanism. For the purpose of comparison a baseline model (multilayer perceptrons) with two hidden layers was also built and tested. The baseline model was trained and tested on all variables excluding those containing the longitudinal information and achieved AUROCs and AUPRCs between 0.70 to 0.81 and 0.21 to 0.44 respectively. The GRU model was trained and evaluated on data till week 28 with all 151 variables, the model outperformed the baseline model and achieved an AUROCs and AUPRCs score ranging between 0.72 to 0.83 and 0.21 to 0.44 respectively. The SHAP value for features and combination of features highlighted the most relevant feature for the models which are as follows Aβ plasma levels, regional brain volumetry, and neurocognitive health.

The article features models showing great potential for the detection of rapid progressors (RPs) patients of Alzheimer’s disease using the baseline and longitudinal data. However, it is fraught with issues around robustness, transparency, and study design. The comments are as follows:

Major comments:

- Model/Code transparency: To ensure robustness it is imperative that the exact framework or libraries were used for building, training and evaluating the models is clearly mentioned in the manuscript. 

- Data: Although the data maybe proprietary and available on request, a smaller sample de-identified dataset must be made available so that researchers can verify and build upon the current findings. Typically "available on reasonable request" ends up being most requests being denied.

- Class imbalance: The proportions of RPs (1) and Not RPs (0) is not clearly mentioned which can help figuring out whether the data is balanced or unbalanced. From Figure 2 it can be seen that AUROC value is quite high whereas AUPRC is low which can be due to unbalanced data. If the data is indeed unbalanced, appropriate measures should be taken to address this 

- Study design: The intervals between observation periods are irregular ranging from 4 weeks to 16 weeks. For predicting RPs, the data till 28th week were used but it might be possible to get the same accuracy with 16 weeks or 20 weeks data, this cannot be checked due to the presence of a huge gap between previous observation periods i.e. week 12 and week 28. If this is an issue with data collection this should be clearly mentioned. 

Minor comments: 

- The title of the article could have been more specific using the deep learning term instead of machine learning 

-For better understanding the model performance, the evaluation metric like confusion matrix and F1 score can also be reported.

- Most of the time (as can be seen in Figure 2) the differences in the accuracy of baseline model (2 layer MLP) and GRU model is not large, which can justify the added advantage of using GRU like model architecture with longitudinal data. It would be important to address if the same accuracy as the GRU model, can be achieved by optimizing baseline MLP model.

- The manuscript Authors could do well by pointing out clear challenges in the analysis with respect to reproducibility on another dataset, potential confounders related to patient genetics beyond the standard markers such as ApoE(as AD is well know to have a genetic component) and how the current study could be improved in the future through models that may capture more variation.

Reviewer #3: 1. With a sample size of 1603, 151 variables seem extreme. Did the authors determine the sample size (DOI: 10.1002/sim.7992)? 

2. The authors mentioned 64 baseline variables (22 demographics + 42 comorbid) and 87 longitudinal variables. I think these 87 longitudinal variables also have baseline measurements. 

3. The definition of the RP needs proper citation. Why were the top 10% of patients with the largest score changes from baseline to week 80 classified as RPs? What's the basis of this 10% cut-off? In my opinion, a better definition would be to define RP as a time-to-event variable, e.g., time from the baseline to RP. The authors clearly ignore the time component, though they have access to rich datasets. 

4. The data-splitting approach as an internal validation is a poor choice. The authors are advised to use better methods, such as cross-validation or bootstrapping (doi:10.1093/eurheartj/ehu207).

5. Since the authors utilized the longitudinal data, I am surprised that the authors report the time-invariant AUC instead of the time-dependent AUC. However, in addition to the AUC, the authors should report the calibration (intercept and slope) and the model's overall performance (e.g., brier score). A decision-curve analysis is also recommended. (doi:10.1093/eurheartj/ehu207)

6. The higher AUC from the model with longitudinal data is expected from a statistical point of view. However, with longitudinal data, it is unclear why the authors did not consider a dynamic prediction model that can be updated over time (DOI: 10.1111/rssa.12611). 

7. More than 30% of missing data seems high. The authors used multiple imputation, which is good. But it is unclear how they evaluate the model performance with multiple imputation. Also, did the authors conduct a sensitivity analysis with a different assumption than missing at random? 

8. It is unclear why the authors dichotomized the age variable. And some categories should be merged due to small frequencies (e.g., race, APOE type). 

9. Overall, whether the models used can select spare a model is unclear. The clinicians may not be interested in using this prediction model because of the large number of predictors with a black box model. There should be a justification for why clinicians could be interested in the model the authors had developed.

10. The authors should report the guideline they followed in developing the prediction model.

Reviewer #4: To all Authors,

I have attached a file containing my impression and the minor considerations I would appreciate your consideration and feedback on.

In the meantime, I thoroughly enjoyed reading your work,

Best regards,

6. PLOS authors have the option to publish the peer review history of their article (what does this mean?). If published, this will include your full peer review and any attached files.

**Do you want your identity to be public for this peer review?** For information about this choice, including consent withdrawal, please see our Privacy Policy.

Reviewer #1: No

Reviewer #2: No

Reviewer #3: No

Reviewer #4: Yes: Simon Gilbert Provost

---

## [Decision Letter · Decision Letter 1]

16 Jan 2024

PDIG-D-23-00271R1

Deep learning to predict rapid progression of Alzheimer’s disease from pooled clinical trials: A retrospective study

PLOS Digital Health

Dear Dr. Ma,

Thank you for submitting your manuscript to PLOS Digital Health. After careful consideration, we feel that it has merit but does not fully meet PLOS Digital Health's publication criteria as it currently stands. Therefore, we invite you to submit a revised version of the manuscript that addresses the points raised during the review process.

Please submit your revised manuscript within 30 days Feb 15 2024 11:59PM. If you will need more time than this to complete your revisions, please reply to this message or contact the journal office at digitalhealth@plos.org. Please include the following items when submitting your revised manuscript:

We look forward to receiving your revised manuscript.

Kind regards,

Hualou Liang

Academic Editor

PLOS Digital Health

Journal Requirements:

Additional Editor Comments (if provided):

Reviewers' comments:

Reviewer's Responses to Questions

**Comments to the Author**

1. If the authors have adequately addressed your comments raised in a previous round of review and you feel that this manuscript is now acceptable for publication, you may indicate that here to bypass the “Comments to the Author” section, enter your conflict of interest statement in the “Confidential to Editor” section, and submit your "Accept" recommendation.

Reviewer #1: (No Response)

Reviewer #3: (No Response)

Reviewer #4: All comments have been addressed

2. Does this manuscript meet PLOS Digital Health’s publication criteria? Is the manuscript technically sound, and do the data support the conclusions? The manuscript must describe methodologically and ethically rigorous research with conclusions that are appropriately drawn based on the data presented.

Reviewer #1: (No Response)

Reviewer #3: Partly

Reviewer #4: Yes

3. Has the statistical analysis been performed appropriately and rigorously?

Reviewer #1: (No Response)

Reviewer #3: No

Reviewer #4: Yes

4. Have the authors made all data underlying the findings in their manuscript fully available (please refer to the Data Availability Statement at the start of the manuscript PDF file)?

Reviewer #1: (No Response)

Reviewer #3: Yes

Reviewer #4: Yes

5. Is the manuscript presented in an intelligible fashion and written in standard English?

Reviewer #1: (No Response)

Reviewer #3: Yes

Reviewer #4: Yes

6. Review Comments to the Author

Reviewer #1: Dear Author,

The rebuttal against the raised queries are satisfactory. Al the queries has been addressed appropriately

Reviewer #3: The authors did an excellent job and significantly improved the manuscript. However, there are still some major issues. Otherwise, the manuscript looks organized. 

1. Small frequency issue: American Indian or Alaska Native, multiple race, APOE type E2/E2 have 1, 4, and 2 frequencies, respectively. Any coefficients for these categories are not interpretable, particularly since the authors used cross-validation to build the model. It was suggested to merge small frequency cells to avoid model convergence issues. 

2. Reporting guideline: Using a checklist/guideline for reporting the study was suggested. Examples include TRIPOD (doi: 10.1186/s12916-014-0241-z), TRIPOD-AI (doi: 10.1136/bmjopen-2020-048008)

3. Calibration: The authors did not evaluate the model in terms of calibration. In addition to the discrimination/accuracy, the authors should report the calibration measure(s). Otherwise, the models are not reliable to use in clinical practice. Calibration is highly important with a large number of predictors with small events per variable as such in this study. Poor calibration can make predictions misleading. (doi: 10.1093/eurheartj/ehu207; 10.1186/s12916-019-1466-7)

4. Combining machine learning with multiple imputation: It was unclear how the authors combined their models or model performance measures with multiple imputed datasets. Doi: 10.1037/met0000478

Reviewer #4: To the Authors:

I earnestly valued the level of attention to detail that was displayed for all answers to all reviewers. However, I would like to draw your attention to the fact that while I am not legitimate to say my view upon the others, at least my remarks were addressed accurately and appropriately. Sincerely appreciate the thoughtful remarks suggesting that you may investigate the possibility of comparing SOTA Deep Learning methods and machine learning-adapted algorithms, among other things. This is an excellent work! I eagerly await its publication.

Best of success and have a joyous Christmas!

Cheers, everyone!

7. PLOS authors have the option to publish the peer review history of their article (what does this mean?). If published, this will include your full peer review and any attached files.

**Do you want your identity to be public for this peer review?** For information about this choice, including consent withdrawal, please see our Privacy Policy. 

Reviewer #1: None

Reviewer #3: No

Reviewer #4: Yes: Simon Gilbert Provost

---

## [Decision Letter · Decision Letter 2]

26 Feb 2024

Deep learning to predict rapid progression of Alzheimer’s disease from pooled clinical trials: A retrospective study

PDIG-D-23-00271R2

Dear Mr Ma,

We are pleased to inform you that your manuscript 'Deep learning to predict rapid progression of Alzheimer’s disease from pooled clinical trials: A retrospective study' has been provisionally accepted for publication in PLOS Digital Health.

Best regards,

Hualou Liang

Academic Editor

PLOS Digital Health

Reviewer Comments (if any, and for reference):

Reviewer's Responses to Questions

**Comments to the Author**

1. If the authors have adequately addressed your comments raised in a previous round of review and you feel that this manuscript is now acceptable for publication, you may indicate that here to bypass the “Comments to the Author” section, enter your conflict of interest statement in the “Confidential to Editor” section, and submit your "Accept" recommendation.

Reviewer #3: All comments have been addressed

2. Does this manuscript meet PLOS Digital Health’s publication criteria? Is the manuscript technically sound, and do the data support the conclusions? The manuscript must describe methodologically and ethically rigorous research with conclusions that are appropriately drawn based on the data presented.

Reviewer #3: Yes

3. Has the statistical analysis been performed appropriately and rigorously?

Reviewer #3: Yes

4. Have the authors made all data underlying the findings in their manuscript fully available (please refer to the Data Availability Statement at the start of the manuscript PDF file)?

Reviewer #3: Yes

5. Is the manuscript presented in an intelligible fashion and written in standard English?

Reviewer #3: Yes

6. Review Comments to the Author

Reviewer #3: The authors have addressed all the concerns. The paper now looks nice, and I recommend accepting it. Congratulations to the authors!

7. PLOS authors have the option to publish the peer review history of their article (what does this mean?). If published, this will include your full peer review and any attached files.

**Do you want your identity to be public for this peer review?** For information about this choice, including consent withdrawal, please see our Privacy Policy.

Reviewer #3: No
